# Will Remotely Based Pulmonary Rehabilitation Water Down Its Effectiveness?

**DOI:** 10.3390/life11111270

**Published:** 2021-11-20

**Authors:** Himanshu Rawal, Sharon D. Cornelison, Sheryl M. Flynn, Jill A. Ohar

**Affiliations:** 1Department of Internal Medicine, Section on Pulmonary Medicine, Critical Care, Allergy and Immunologic Diseases, Atrium Health Wake Forest Baptist, Medical Center Boulevard, Winston-Salem, NC 27157, USA; hrawal@wakehealth.edu; 2Department of Pulmonary and Cardiac Rehabilitation, J. Paul Sticht Center on Aging and Rehabilitation, Atrium Health Wake Forest Baptist, Medical Center Boulevard, Winston-Salem, NC 27157, USA; scorneli@wakehealth.edu; 3Blue Marble Health, Lincoln Avenue, Altadena, CA 91001, USA; sheryl@bluemarblehealthco.com

**Keywords:** remotely, tele-rehabilitation, pulmonary, COPD

## Abstract

Despite numerous benefits, traditional Pulmonary Rehabilitation (PR) as a resource remains underutilized in chronic lung disease. Less than 3% of eligible candidates for PR attend one or more sessions after hospitalization due to many barriers, including the ongoing COVID-19 pandemic. Emerging alternative models of PR delivery such as home-based PR, tele-rehabilitation, web-based PR, or hybrid models could help address these barriers. Numerous studies have tested the feasibility, safety, and efficacy of these methods, but there is wide variability across studies and methods. We conducted a literature review to help determine if these alternative delivery methods watered down the effectiveness of PR. To evaluate the effectiveness of remotely based PR, the authors performed a literature search for randomized controlled trials (RCTs), cohort studies, and case series using PubMed, CINAHL, and Medline to identify relevant articles through 1 May 2021. Twenty-six applicable studies were found in which 11 compared tele-rehabilitation to conventional clinic-based PR; 11 evaluated tele-rehabilitation using the patient’s baseline status as control; and four compared tele-rehabilitation to no rehabilitation. Despite the different technologies used across studies, tele-rehabilitation was found to be both a feasible and an efficacious option for select patients with lung disease. Outcomes across these studies demonstrated similar benefits to traditional PR programs. Thus the existing data does not show that remotely based PR waters down the effectiveness of conventional PR. Use of remotely based PR is a feasible and effective option to deliver PR, especially for patients with significant barriers to conventional clinic-based PR. Additional, well-conducted RCTs are needed to answer the questions regarding its efficacy, safety, cost-effectiveness and who, among patients with COPD and other lung diseases, will derive the maximum benefit.

## 1. Introduction

Pulmonary rehabilitation (PR) is a “comprehensive intervention based on a thorough patient assessment followed by patient-tailored therapies that include, but are not limited to, exercise training, education, and behavior change, designed to improve the physical and psychological condition of people with chronic respiratory disease and to promote the long-term adherence to health-enhancing behaviors” [1]. It is an integral part of the management of patients with various respiratory diseases, along with other well-established modalities (e.g., oxygen, inhalers, and non-invasive ventilation). PR improves exercise capacity [1,2,3], limb muscle function [1,2,4], and health-related quality of life [1,2,5]. PR reduces symptom burden [1,2,6], healthcare costs [1,2], hospitalizations [1,2], and other unplanned health care utilization [1,2]. Moreover, it has been shown to decrease mortality [7], as well as anxiety and depression [1,2]. Despite numerous benefits, PR as a resource remains underutilized. Less than 3% of eligible candidates for PR attend one or more sessions after hospitalization [8,9,10]. Several barriers affect PR attendance and adherence. Enthusiasm of the referring physician, travel and distance to the PR center, caregiver knowledge and availability, smoking history, lack of trained manpower, and suboptimal funding are some of the most common barriers [1,11]. In a systematic review by Young et al., travel and transport were consistently reported as barriers across studies [12]. The current COVID-19 pandemic has significantly impacted clinic-based PR program enrollment and attendance, as people with chronic respiratory diseases are at an increased risk of severe COVID-19 disease [13], and many have been advised to stay at home and avoid in person contacts. This recommendation has led to a further widening in the already existing gap in care for those living with respiratory diseases. Emerging alternative models of PR delivery such as home-based PR, tele-rehabilitation, web-based PR, or hybrid models could help address these barriers and close the gap. In 2015, the American Thoracic Society (ATS) and European Respiratory Society (ERS) recommended further research to develop alternative delivery models to help improve access to PR. The ATS/ERS also stated that “adoption of alternative models for PR will require demonstration of comparable or greater clinical outcomes to those of traditional PR programs, as well as evaluation of safety and cost-effectiveness, staff training and guideline development” [1]. Numerous studies have tested the feasibility, safety, and efficacy of these methods, but there is wide variability across studies and methods [14]. In this article, we reviewed studies to inform the question “Will remotely-based PR water down its effectiveness?”

## 2. Methods

We performed a 10-year literature review using PubMed, CINAHL, and Medline to identify relevant articles through 1 May 2021, for studies evaluating effectiveness of remotely-based PR. We reasoned that publications prior to 2011 would likely be less applicable, from a technology standpoint. The search strategy involved the keywords: tele-rehabilitation; pulmonary rehabilitation; rehabilitation; telehealth; chronic obstructive pulmonary disease (COPD); chronic lung disease; and COVID-19. The search was supplemented with articles from our personal files. We sought to identify Randomized Control Trials (RCTs), cohort studies, and case series. A total of 26 studies were identified, 10 of which were RCTs, 15 were cohort studies, and one was a case series (Table 1) (Figure 1).

## 3. Results

Out of the total 26 studies conducted between 2011 and 2021; 15 (58%) were cohort studies, 10 (38%) were RCTs and one (4%) was a case series (Table 1). Twenty-one (81%) of the studies included participants with COPD [16,17,18,19,20,21,22,23,24,25,26,29,30,31,32,33,34,35,36,39,40], four (15%) with COVID-19 [27,28,37,38] and one (4%) with cystic fibrosis [15].

### 3.1. Tele-Rehabilitation vs. Conventional Pulmonary Rehab

Eleven studies compared tele-rehabilitation to conventional clinic-based PR (Table 2), seven of which were RCTs [16,17,18,19,20,21,22] and 4 were cohort studies [15,23,24,25]. The RCTs included a total of *n =* 979 study participants with *n =* 477 participants receiving traditional PR while 502 received a tele-rehabilitation-based intervention.

Modes of Tele-rehabilitation: A variety of modes of tele-rehabilitation were used, including real-time broadcast by a healthcare provider (HCP) to the participant’s home [16,23] and to a satellite center near the participant’s home [25]. These were standardized group sessions with durations varying from 30 min [16], to 60 min [23] and to 2 h [25]. Others used an online platform with pre-recorded sessions [20,21,22,24] or a smartphone-based application [15]. The sessions were individualized and were performed at the patients’ convenience. Patients’ progress was monitored online using questionnaires [15,20,21,24]. In all the studies there was contact with the research team or HCP via email or phone to ensure adequate patient progress or to answer queries [15,20,21,22,24]. Unsupervised home-based rehabilitation with standardized weekly phone calls by a HCP to monitor patient progress was used in two studies [18,19]. A virtual gaming system with and without conventional PR was used in one study [17].

Outcomes: Outcomes from these studies found that, compared with conventional clinic-based PR, the tele-rehabilitation group demonstrated the following improvements: better adherence [15], upper extremity strength as measured by the arm curl test [17], leg strength as measured by the Chair Stand Test [17], functional behavior as measured by the Up and Go Test [17] and physical activity (steps per day) [23]. Distance outcomes from the 6 min walk test (6MWT) varied from increased [17,18], not different [16,23], and not inferior to conventional clinic-based PR [21]. Dyspnea as measured by the Chronic Respiratory Questionnaire (CRQ) was not different in one study [22] when compared to conventional PR; however, in another study, it was inferior [19]. Improvement was seen in the Endurance Shuttle Walk test (ESWT) and were not different from improvements seen in traditional PR [22]. Outcomes from the Modified Medical Research Council (mMRC) were improved [24] and not different [23] from those found with conventional PR. When tele-rehabilitation was compared to conventional PR, St. George’s Respiratory Questionnaire (SGRQ) improved in both groups [25], and the differences between groups were not different [23,25]. When compared to pharmacotherapy alone, both home-based PR and conventional PR decreased COPD exacerbations and hospitalization [25].

### 3.2. Tele-Rehabilitation Only (Pre-Intervention vs. Post Intervention)

Eleven studies evaluated tele-rehabilitation using the patient’s baseline status as control (Table 3), ten of which were cohort studies [26,27,29,30,31,32,33,34,35,36] and one was a case series [28]. Seven studies were both efficacy and feasibility trials, whereas four of them evaluated feasibility alone [28,34,35,36]. A total of *n* = 187 participants were included in these studies.

Modes of Tele-Rehabilitation: Studies involving tele-rehabilitation without a comparison group used a variety of interventional modes including real-time broadcast by an HCP to the participant’s home [26,30,32,34,36]. Zanaboni et al. [30] used an individualized treadmill program whereas standardized group sessions were used in other studies [26,32,34,36]. Unsupervised pre-recorded videos on a smartphone application and online platform were used in two studies [29,31]. Patients in these two studies were expected to participate daily in one study [29] and three times a week in another [31]. Virtual gaming was used in two studies [33,35]. One study used virtual gaming in addition to conventional PR [35] three times a week, whereas a virtual gaming system alone was used in the other study [33]. Unsupervised weekly phone calls to remotely monitor patient progress were used in two studies [27,28].

Outcomes: All 11 studies found that tele-rehabilitation was feasible [26,27,28,29,30,31,32,33,34,35,36] (Table 3). When comparing study endpoints before and after the intervention, the tele-rehabilitation group had better adherence (defined as > 50% sessions completed) [35], improved upper extremity strength as measured by the arm lift test [33], and increased leg strength as measured by the Sit to Stand Test [26,27,28,33]. There was also an improvement in both the 6MWT [27,30,32,34,36] and ESWT [33]. Dyspnea, as measured by the CRQ [26,32,34] and CAT scores [29,30], were improved after tele-rehabilitation. There was also an improvement in Health-Related Quality of Life (HRQoL) using the CRQ scores [26,29,33]. Hoaas et al. (2016), examined the provision of equipment for self-management and unsupervised home exercise for 1 year after an intervention. They found a decrease in physical activity (steps per day) suggesting that the availability of equipment might not be sufficient to maintain physical activity levels post-rehab completion [31].

### 3.3. Tele-Rehabilitation vs. No Rehabilitation

Four studies (Table 4) compared tele-rehabilitation to no rehabilitation [37,38,39,40], three of which were RCTs [37,38,40], and one was a cohort study [39]. The RCTs included a total of *n* = 234 participants, of which *n* = 115 received a tele-rehabilitation-based intervention. The lone cohort study was both a feasibility and an efficacy study [39].

Modes of Tele-Rehabilitation: When comparing tele-rehabilitation to no rehabilitation, the various modes of delivery included real-time broadcast by an HCP to the participant’s home [39,40], pre-recorded sessions on a smartphone-based application [38], and twice weekly phone calls to otherwise unsupervised study participants [37]. Among studies utilizing real time broadcast, one study [39] used individualized sessions three times a week for 12 weeks, whereas group sessions three times a week for 8 weeks was used in the other [40]. Another study used a smartphone-based application which provided weekly teleconsultations in addition to 3–4 unsupervised sessions per week [38]. Unsupervised breathing exercises for 7 days with biweekly check-ins using telephone calls was used in one study [37].

Outcomes: Tele-rehabilitation compared with no rehabilitation was found to be feasible [37,39] and efficacious [38,40]. These studies showed that compared to no rehabilitation intervention, the tele-rehabilitation group showed improvements in the following areas: perceived dyspnea using the Borg scale [37], lower limb muscle strength using a static squat test [38], HRQoL using Short Form Health Survey-12 (SF-12), Physical Component Score (PCS) and CRQ scores, [38,40], and both the 6MWT and ESWT [37,38,40]. Psychological benefit was also seen using the Hospital Anxiety and Depression Scale (HADS) score [40]. Decreased mortality and readmission rates due to COPD exacerbations were also seen in this group [39].

## 4. Discussion

Remotely-based PR is a feasible option for people living with chronic lung diseases, especially those with COPD [16,17,18,19,20,21,22,23,24,25,26,29,30,31,32,33,34,35,36,39,40]. It may also be an option for patients recovering from the remote effects of COVID-19 [27,28,37,38]. Despite the different technologies used for telecommunication, these studies support tele-rehabilitation as a feasible option. In all the studies, both an educational component and a physical activity component was present [15,16,17,18,19,20,21,22,23,24,25,26,27,28,29,30,31,32,33,34,35,36,37,38,39,40].

When compared to no rehabilitation, tele-rehabilitation was effective across studies. It consistently showed statistically significant improvements in exercise capacity [17,18,22,26,27,28,30,32,33,34,36,37,38,40] and dyspnea perception [19,22,24,25,26,32,34,37,38,40] as well as a decreased 30-day mortality and readmission rates for acute exacerbations due to COPD [39]. This makes tele-rehabilitation an attractive option for patients who have significant barriers to attending conventional, clinic-based PR. When compared to conventional PR, tele-rehabilitation was found to be equally effective in the majority of studies. Multiple non-inferiority trials showed no significant difference in outcomes between groups, and instead found tele-rehabilitation to be as effective as conventional PR [19,21]. Hansen et al. (2020), failed to show that telehealth was superior to conventional PR, however participants in the tele-rehabilitation group had higher completion rates [16].

Despite several studies showing feasibility and non-inferiority to conventional PR, the acceptance and implementation of remotely based tele-rehabilitation has been slow and highly variable over the years [14]. The current COVID-19 pandemic has greatly impacted conventional, in-person rehabilitation enrollment and attendance. This has re-focused the spotlight on tele-rehabilitation as an option for patients with chronic lung disease. However, the studies are heterogeneous in design with small sample sizes, use inconsistent outcome measures, and most importantly, use a wide variety of technologies [41]. Another major hurdle that has likely reduced the uptake of remote-based PR is the paucity of data regarding its cost effectiveness. Tele-rehabilitation involves using complex technology and equipment to monitor patients, and it also needs trained manpower—which can be expensive. Many insurers are willing to pay for conventional PR but not tele-rehabilitation [42]. Wide-spread acceptance of tele-rehabilitation in the healthcare system across the US is hampered by regulations and restrictions by state governments and policies of insurers [43]. Limitations such as potential for injury, digital and health literacy, and lack of appropriate device/internet connection remain. It is important to understand when tele-rehabilitation is safe and which subtype of patient benefits the most from tele-rehabilitation. Small studies have shown that tele-rehabilitation is safe with no major adverse events, [15,18,19,21,23,26,27,34,35,37,39], but large studies are lacking. The majority of the literature focuses on patients with COPD, thus making it difficult to extrapolate benefits to other lung diseases [44]. Challenges with digital literacy and familiarity are encountered especially in elderly frail patients. In a study by Chaplin et al. [22], high dropout rates were seen in the tele-rehabilitation group with technological challenges being the major reason. Investigators were required to modify their platform according to patient feedback.

## 5. Conclusions

The existing data does not show that remotely based PR will water down the effectiveness of conventional PR. Use of remotely based PR is a feasible and effective option to deliver PR, especially for patients with significant barriers to conventional clinic-based PR. Additional, well-conducted RCTs are needed to answer the questions regarding its efficacy, safety, cost-effectiveness and who will derive the maximum benefit among patients with COPD and other lung diseases.

## Figures and Tables

**Figure 1 life-11-01270-f001:**
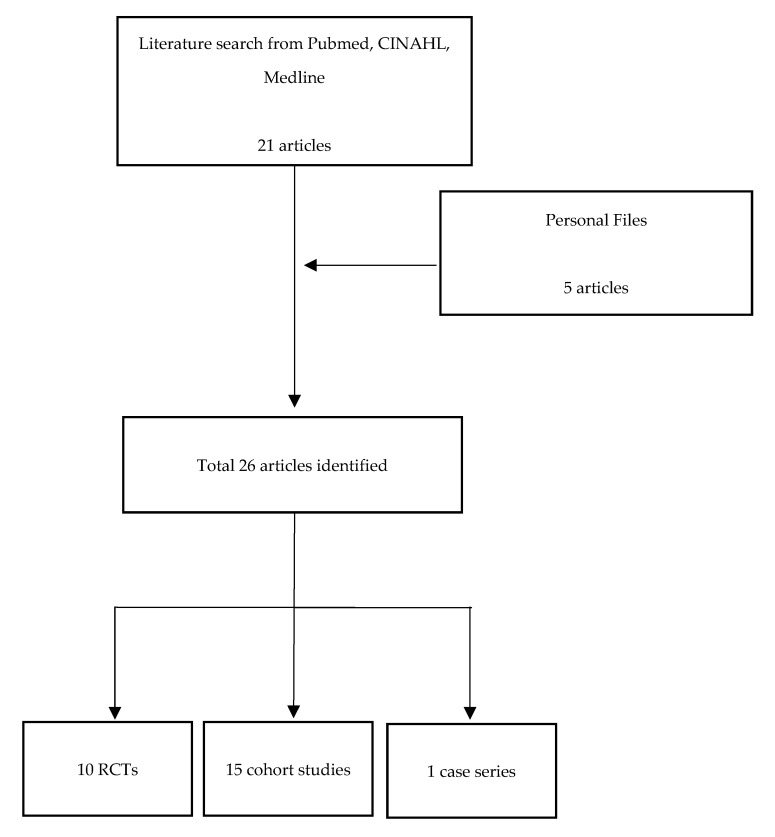
Flow diagram illustrating selection of eligible studies.

**Table 1 life-11-01270-t001:** Studies evaluating tele-rehabilitation.

Citation	Study Design/Purpose	Patient Disease/Sample Size	Rehab Site	Intervention	Results	Adverse Events
Tele-Rehabilitation vs. Conventional Pulmonary Rehab						
Layton et al., 2021 [15]	Cohort Study/ Feasibility and efficacy Study	Cystic Fibrosis/ *n* = 11 vs. *n* = 8	Home	Smartphone based application	Increased adherence (*p* = 0.03)	Muscle pain
Hansen et al., 2020 [16]	RCT/ Superiority trial	COPD/ *n* = 67 vs. 67	Home	Real time broadcast by physiotherapist and nurse	No difference in 6MWT between groups. Higher rate of completion in the tele rehabilitation group (*p* < 0.01)	2 AE’s in the Conventional PR group.
Rutkowski et al., 2020 [17]	RCT/ Superiority trial	COPD/ *n* = 106 *n* = 34 Conventional PR *n* = 38 Conventional PR +VR *n* = 34 VR	Inpatient	Virtual game system Conventional PR with physiotherapist	PR+VR group was superior to just PR group in Arm curl (*p* < 0.003), chair stand (*p* < 0.008), Up and Go (*p* < 0.000), and 6MWT (*p* < 0.011)VR group superior to PR in Arm curl (*p* < 0.000), chair stand (*p* < 0.001) and 6MWT (*p* < 0.031)	Not mentioned ^*^
Bernocchi et al., 2018 [18]	RCT/Feasibility and efficacy Study	COPD/ *n* = 56 vs. *n* = 56	Home	Physiotherapist weekly phone call	Improvement in 6MWT (*p* < 0.0040)	None reported ^**^
Horton et al., 2018 [19]	RCT/ Non-inferiority trial	COPD/ *n* = 145 vs. *n* = 142	Home	Physiotherapist weekly phone call	No improvement in CRQ dyspnea (*p* = 0.18)	None reported
Vasilopoulou et al., 2017 [20]	RCT/Efficacy Trial	COPD/ *n* = 47 vs. *n* = 50	Home	Recorded session by physiotherapist, exercise scientist	Home based PR and Conventional PR decreased COPD exacerbation and hospitalization when compared to pharmacotherapy (*p* < 0.001)	Not mentioned
Bourne et al., 2017 [21]	RCT/ Non-inferiority trial	COPD/ *n* = 64 vs. *n* = 26	Home	Pre-recorded session by a physiotherapist	Online PR was non-inferior to Conventional PR in 6MWT (*p* = 0.098) and CAT (*p* = 0.373)	Back pain and muscular pain
Chaplin et al., 2017 [22]	RCT/Feasibility and efficacy trial	COPD/ *n* = 51 vs. *n* = 52	Home	Pre-recorded session physiotherapist	No significant difference in the CRQ dyspnea (*p* > 0.05), ESWT (*p* > 0.05)	Not mentioned
Paneroni et al., 2015 [23]	Cohort Study/ Feasibility and Efficacy Study	COPD/ *n* = 18 vs. *n* = 18	Home	Real time video conferencing with the physiotherapist	Improvement in physical activity (steps per day) (*p* < 0.0002) No statistically significant difference in 6MWT, SGRQ or mMRC	None reported
Tabak et al., 2014 [24]	Cohort Study/ Feasibility and Efficacy Study	COPD/*n* = 15 vs. *n* = 14	Home	Pre-recorded session by physiotherapist	Improvement in mMRC scale (*p* < 0.03)	Not mentioned
Stickland et al., 2011 [25]	Cohort Study/ Efficacy Study	COPD/ *n* = 147 vs. *n* = 262	Satellite Center under supervision	Real time video conferencing with respiratory therapist	Both telehealth PR and Conventional PR showed improvement in SGRQ (*p* < 0.05)	Not mentioned
**Tele-rehabilitation Alone (Pre vs. Post Intervention)**						
Lewis et al., 2021 [26]	Cohort Study/ Efficacy and Feasibility Study	COPD/ *n* = 17	Home	Physiotherapist by real time video conferencing	Improvements in 1 min STS (*p* = 0.004), GAD (*p* = 0.023), PHQ-9 (*p* = 0.029), CRQ dyspnea (*p* = 0.001), CRQ fatigue (*p* = 0.004), CRQ emotion (*p* = 0.0002), CRQ mastery (*p* = 0.001)	None reported
Paneroni et al., 2021 [27]	Cohort Study/ Efficacy and Feasibility Study	COVID-19/ *n* = 25	Home	Twice a week call by a physiotherapist	Improvement in STS (*p* = 0.003) and 6MWT (*p* = 0.0006)	None Reported
Wootton et al., 2020 [28]	Case Series	COVID-19/ *n* = 3	Home	Weekly call by physiotherapist	Improvement in 1 min and 5 min STS	Not mentioned
Rassouli et al., 2018 [29]	Cohort Study/Efficacy and Feasibility Study	COPD/*n* = 34	Home	Smartphone application; pre-recorded videos	Improvement in CAT scores (*p* = 0.008) Improvement in CRQ fatigue (*p* < 0.001), mastery (*p* < 0.001) and emotion (*p* < 0.001).	Not mentioned
Zanaboni et al., 2017 [30]	Cohort Study/ Efficacy and Feasibility study	COPD/ *n* = 10	Home	Real time video conferencing with Physiotherapist	Improvement in 6MWT, CAT (*p* = 0.022) scores	Not mentioned
Hoaas et al., 2016 [31]	Cohort Study/ Efficacy and Feasibility Study	COPD/*n* = 10	Home	Pre-recorded session by physiotherapist	Decrease in physical activity (Steps per day) 1 year after a 2-year intervention (*p* = 0.039)	Not mentioned
Marquis et al., 2014 [32]	Cohort Study/ Efficacy and Feasibility Study	COPD/ *n* = 26	Home	Combined Real-time video conferencing by physiotherapists and unsupervised sessions	Improvement in 6MWT (*p* < 0.001), CET (*p* = 0.003) and CRQ (*p* < 0.001) at 8 weeks but not sustained until 24-week follow-up	Not mentioned
Albores et al., 2013 [33]	Cohort Study/ Efficacy and Feasibility Study	COPD/ *n* = 25	Home	Virtual Game system	Improvement in ESWT (*p* = 0.005), arm-lift (*p* = 0.03), sit to stand repetitions (*p* = 0.03) and CRQ emotion scores (*p* = 0.02)	Not mentioned
Holland et al., 2013 [34]	Cohort Study/ Feasibility Study	COPD/ *n* = 8	Home	Real-time videoconferencing with physiotherapist	Improvement in 6MWT, CRQ score	Minor adverse events were desaturation < 88% (*n*=1) & heart rate >150 BPM(*n*=1)
Wardini et al., 2013 [35]	Cohort Study/ Feasibility Study	COPD/ *n* = 32	Inpatient conventional + virtual	Virtual game system	Increased enjoyment using VAS Increased adherence	None reported
Tousignant et al., 2012 [36]	Cohort Study/ Feasibility Study	COPD/*n* = 3	Home	Real time videoconferencing with physiotherapist	Improvement in 6MWT for 2 out of 3 participants	Not mentioned
**Tele-rehabilitation vs. No Rehabilitation**						
Gonzalez-Gerez et al., 2021 [37]	RCT/ Feasibility and Efficacy Trial	COVID-19/ *n* = 19 vs. *n* = 19	Home	Twice weekly calls by physiotherapist	Improvement in 6MWT (*p* < 0.001) and dyspnea perception using Borg scale (*p* < 0.001)	None Reported
Li et al., 2021 [38]	RCT/ Efficacy Trial	COVID-19/ *n* = 59 vs. *n* = 61	Home	Smartphone-based application	Improvement in 6MWT (*p* < 0.001), mMRC (*p* < 0.001), LMS (*p* < 0.001) and SF-12 PCS (*p* < 0.001)	None reported
Bhatt et al., 2019 [39]	Cohort Study/ Feasibility and Efficacy Study	COPD/ *n* = 80 vs. *n* = 160	Home	Physiotherapist by real-time video conferencing	Decreased 30-day all-cause mortality (*p* = 0.013) and readmissions due to AECOPD (*p* = 0.04)	None reported
Tsai, 2017 [40]	RCT/ Efficacy Trial	COPD/ *n* = 37 vs. *n* = 37	Home	Real-time broadcast by physiotherapist	Improvement in ESWT (*p* < 0.001), self-efficacy (*p* < 0.007) and CRQ (*p* = 0.07)	Not mentioned

AE: Adverse Event; AECOPD: Acute Exacerbation of Chronic Obstructive Pulmonary Disease; CAT: COPD Assessment Test; CET: Constant work rate Exercise Test; COPD: Chronic Obstructive Pulmonary Disease; CRQ: Chronic Respiratory Questionnaire; ESWT: Endurance Shuttle Walk Test; EQ-VAS: EuroQol Visual Analog Scale; GAD: Generalized Anxiety Disorder; MRC: Medical Research Council; mMRC: Modified Medical Research Council; LMS: Lower limb muscle Strength; PCS: Physical Component Score; PHQ-9: Primary Health Questionnaire-9; PR: Pulmonary Rehabilitation; RCT: Randomized Control Trial; SF-12: Short Form Health Survey-12; SGRQ: St George’s Respiratory Questionnaire; STS: Sit To Stand; VR: Virtual Reality; 6MWT: 6 Minute Walk Test. * Not mentioned—Studies did not look for adverse events. ** None reported—Studies reported the absence of adverse events.

**Table 2 life-11-01270-t002:** Tele-rehabilitation vs. Conventional Pulmonary Rehabilitation.

Outcome	Improved ^a^	Inferior ^b^	Not Different ^c^	Not Inferior ^d^
6MWT	Rutkowski et al. [17], Bernocchi et al. [18]		Hansen et al. [16], Paneroni et al. [23]	Bourne et al. [21]
CRQ dyspnea		Horton et al. [19]	Chaplin et al. [22]	
ESWT			Chaplin et al. [22]	
mMRC	Tabak et al. [24]		Paneroni et al. [23]	
SGRQ	Stickland et al. [25] —so did traditional		Paneroni et al. [23]	
CAT				Bourne et al. [21]
Arm Curl	Rutkowski et al. [17]			
Chair Stand	Rutkowski et al. [17]			
Up and Go	Rutkowski et al. [17]			
Physical Activity (steps per day)	Paneroni et al. [23]			

CRQ: Chronic Respiratory Disease; ESWT: Endurance Shuttle Walk Test; CAT: COPD Assessment Test; HADS: Hospital Anxiety and Depression Scale; mMRC: Modified Medical Research Council; SGRQ: St. George’s Respiratory Questionnaire; 6MWT: 6 Minute Walk Test. **^a^** Statistically significant improvement found in both groups. **^b^** In this non-inferiority study tele-rehab was found to be inferior to conventional PR. **^c^** No statistical difference in outcomes between intervention and control group. **^d^** Non inferiority threshold reached in the non-inferiority RCT.

**Table 3 life-11-01270-t003:** Tele-rehabilitation (Pre vs. Post Intervention).

Outcome	Improved	No Improvement
6MWT	Paneroni et al. [27] Zanaboni et al. [30] Marquis et al. [32] Holland et al. [34] Tousignant et al. [36]	
CRQ dyspnea	Marquis et al. [32] Holland et al. [34] Lewis et al. [26]	
ESWT	Albores et al. [33]	
CAT	Zanaboni et al. [30] Rassouli et al. [29]	
Arm Curl	Albores et al. [33]	
Chair Stand	Lewis et al. [26] Paneroni et al. [27] Wootton et al. [28] Albores et al. [33]	
Physical Activity (steps per day)		Hoaas et al. [31]

CRQ: Chronic Respiratory Questionnaire; ESWT: Endurance Shuttle Walk Test; CAT: COPD Assessment Test; 6MWT: 6 Minute Walk Test.

**Table 4 life-11-01270-t004:** Tele-rehabilitation vs. No Rehabilitation.

Outcome	Improved	Not Different
6MWT	Gonzalez-Gerez et al. [37] Li et al. [38]	Tsai et al. [40]
Borg dyspnea scale	Gonzalez-Gerez et al. [37]	
ESWT	Tsai et al. [40]	
mMRC	Li et al. [38] Tsai et al. [40]	
Static Squat Test	Li et al. [38]	
HADS	Tsai et al. [40]	
CRQ	Tsai et al. [40]	
30-day all-cause mortality	Bhatt et al. [39]	

CRQ: Chronic Respiratory Questionnaire; ESWT: Endurance Shuttle Walk Test, CAT: COPD Assessment Test; HADS: Hospital Anxiety and Depression Scale; mMRC: Modified Medical Research Council; 6MWT: 6 Minute Walk Test.

## Data Availability

Not applicable.

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
