# Peer review of "Will Remotely Based Pulmonary Rehabilitation Water Down Its Effectiveness?"

_life, 2021, doi:10.3390/life11111270_

Round 1
Reviewer 1 Report
Methods - could you give the n number for the number of studies from your own personal files and the number identified from the search. If possible could a PRISMA flowchart be included
Did all of the interventions include both an education and physical activity component. Some more detail on the interventions is needed to help compare the similarities and differences between the studies. Some detail needed on the limitations for some undertaking tele-rehabilitation - i.e. potential for injury, impact of digital and health literacy, lack of appropriate device/internet connection. do any of the studies report patient feedback?
Author Response
Regarding reviewer 1’s comments:
- Methods - could you give the n number for the number of studies from your own personal files and the number identified from the search? If possible, could a PRISMA flowchart be included
Response:
We have added the PRISMA flowchart which includes both personal files and literature searched studies. (Line 75-106)
- Did all of the interventions include both an education and physical activity component?
Response:
We have added that educational and physical activity component was part of all the studies. [Line 230-231]
It now reads as “In all the studies, both an educational component and a physical activity component was present. [15-40]”
- Some more detail on the interventions is needed to help compare the similarities and differences between the studies.
Response:
We have added a brief detail of different modes used for telerehabilitation under each heading (conventional PR vs tele-rehabilitation; tele-rehabilitation only; tele-rehabilitation vs no rehabilitation) [Line 131-140; 171-180; 201-210]
Line 131-140 reads as “These were standardized group sessions with duration varying from 30 minutes [16], 60 minutes [23] and 2 hours [25]. Others used an online platform with pre-recorded sessions [20-22, 24] or a smartphone-based application [15]. The sessions were individualized and were performed at patient’s convenience. Patient’s progress was monitored online using questionnaires [15, 20-21, 24]. In all the studies there was contact with the research team or HCP via email or phone to ensure adequate patient progress or to answer queries. [15, 20-22, 24]. Unsupervised home-based rehabilitation with standardized weekly phone calls by a HCP to monitor patient progress was used in two studies [18, 19]. A virtual gaming system with and without conventional PR was used in one study [17].”
Line 171-180 reads as “Studies involving tele-rehabilitation without a comparison group used a variety of interventional modes including real-time broadcast by an HCP to the participant’s home [26, 30, 32, 34, 36]. Zanaboni et al [30] used an individualized treadmill program whereas standardized group sessions were used in other studies [26, 32, 34, 36]. Unsupervised pre-recorded videos on a smartphone application and online platform were used in two studies [29, 31]. Patients in these two studies were expected to participate daily in one study [29] and three times a week in other [31]. Virtual gaming was used in 2 studies [33, 35]. One study used virtual gaming in addition to conventional PR [35] three times a week whereas, only virtual gaming system was used in the other study [33]. Unsupervised weekly phone calls to remotely monitor patient progress were used in 2 studies [27, 28].”
Line 201-210 reads as “ When comparing tele-rehabilitation to no rehabilitation, the various modes of delivery included real-time broadcast by an HCP to the participant’s home [39, 40], pre-recorded sessions on a smartphone-based application [38], and twice weekly phone calls to otherwise unsupervised study participants [37]. Among studies utilizing real time broadcast, one study [39] used individualized sessions 3 times a week for 12 weeks, whereas group sessions 3 times a week for 8 weeks was used in the other [40]. Another study used a smartphone based application which provided once a week teleconsultations in addition to 3-4 unsupervised sessions per week [38]. Unsupervised breathing exercises for 7 days with biweekly check-ins using telephone calls was used in one study [37].”
- Some detail needed on the limitations for some undertaking tele-rehabilitation - i.e. potential for injury, impact of digital and health literacy, lack of appropriate device/internet connection. do any of the studies report patient feedback?
Response:
Thank you for the suggestion. We have added the safety aspect, and other potential limitations that can come with telerehabilitation.[Line 255-265]
Line 255-265 reads as” Limitations like potential for injury, digital and health literacy, lack of appropriate device/internet connection remain. It is important to understand when tele-rehabilitation is safe and which subtype of patient benefits the most from tele-rehabilitation. Small studies have shown that tele-rehabilitation is safe with no major adverse events, [15,18,19,21,23,26,27,34,35,37,39], but large studies are lacking. The majority of the literature focuses on patients with COPD, thus making it difficult to extrapolate benefits to other lung diseases [44]. Challenges with digital literacy and familiarity are encountered especially in elderly frail patients. In a study by Chaplin et al [22], high dropout rates were seen in the tele-rehabilitation group with technological challenges being the major reason. Investigators were required to modify their platform according to patient feedback.”

Reviewer 2 Report
Rawal Himanshu et al. proposed a literature review, investigating a timely subject: remotely based pulmonary rehabilitation effectiveness for patients with a chronic pulmonary disease.
The manuscript is well written, with adapted references, and an appropriate design/method. The introduction well present objective of the review and current limitation for convention pulmonary rehabilitation.
While most of studies included COPD patients, authors manage to include COVID-19 related studies.
Authors well balanced their discussion though various features: tele-rehabilitation vs conventional PR, tele-PR alone, tele-PR vs no intervention.
3 tables provide fast and relevant overview of studies by categories, all presented with their respective results and limitations.
The MS can be considered in its current form.
Thanks to the authors and the editorial team for this review request.
Author Response
Regarding Reviewer 2’s comments –
- Rawal Himanshu et al. proposed a literature review, investigating a timely subject: remotely based pulmonary rehabilitation effectiveness for patients with a chronic pulmonary disease.
- The manuscript is well written, with adapted references, and an appropriate design/method. The introduction well present objective of the review and current limitation for convention pulmonary rehabilitation.
- While most of studies included COPD patients, authors manage to include COVID-19 related studies.
- Authors well-balanced their discussion though various features: tele-rehabilitation vs conventional PR, tele-PR alone, tele-PR vs no intervention.
- 3 tables provide fast and relevant overview of studies by categories, all presented with their respective results and limitations.
- The MS can be considered in its current form.
- Thanks to the authors and the editorial team for this review request.
